

# Evolution of fertilization ability in obligatorily outcrossing populations of *Caenorhabditis elegans*

Joanna K. Palka, Alicja Dyba, Julia Brzozowska, Weronika Antoł, Karolina Sychta and Zofia M. Prokop

Institute of Environmental Sciences, Faculty of Biology, Jagiellonian University in Cracow, Cracow, Poland

## ABSTRACT

In species reproducing by selfing, the traits connected with outcrossing typically undergo degeneration, a phenomenon called selfing syndrome. In *Caenorhabditis elegans* nematodes, selfing syndrome affects many traits involved in mating, rendering cross-fertilization highly inefficient. In this study, we investigated the evolution of cross-fertilization efficiency in populations genetically modified to reproduce by obligatory outcrossing. Following the genetic modification, replicate obligatorily outcrossing were maintained for over 100 generations, at either optimal (20 °C) or elevated (24 °C) temperatures, as a part of a broader experimental evolution program. Subsequently, fertilization rates were assayed in the evolving populations, as well as their ancestors who had the obligatory outcrossing introduced but did not go through experimental evolution. Fertilization effectivity was measured by tracking the fractions of fertilized females in age-synchronized populations, through 8 h since reaching adulthood. In order to check the robustness of our measurements, each evolving population was assayed in two or three independent replicate blocks. Indeed, we found high levels of among-block variability in the fertilization trajectories, and in the estimates of divergence between evolving populations and their ancestors. We also identified five populations which appear to have evolved increased fertilization efficiency, relative to their ancestors. However, due to the abovementioned high variability, this set of populations should be treated as candidate, with further replications needed to either confirm or disprove their divergence from ancestors. Furthermore, we also discuss additional observations we have made concerning fertilization trajectories.

# INTRODUCTION

In the animal kingdom, sexual reproduction is predominant and mating systems vary in their stunning diversity. In most animal taxa, individuals need to combine carriers of genetic material—gametes—with those of another individual. This form of reproduction is called outcrossing. Less frequently, animals fuse gametes within one individual, in a process called self-fertilization or selfing. Transitions from outcrossing to selfing have occurred repeatedly during evolution (*e.g.*, *Barrett, 2008*; *Jarne & Auld, 2006*). Such transition tends to affect numerous organismal traits, including, in particular, degeneration of those traits

Corresponding author
Joanna K. Palka,
joanna.palka22@gmail.com

connected with cross-fertilization. This phenomenon is called selfing syndrome (*Cutter, 2008*; *Shimizu & Tsuchimatsu, 2015*).

In the nematode genus *Caenorhabditis*, the transition from obligatory dioecious (male–female) outcrossing to androdioecy (with selfing hermaphrodites predominating in populations and only occasionally outcrossing with rare males) happened at least three times independently (*Kiontke et al., 2004*; *Thomas, Woodruff & Haag, 2012*). One of the species that underwent the reproductive mode transition is *Caenorhabditis elegans*. In this species, males usually constitute <0.5% of populations, and selfing syndrome is visible in traits of both sexes (hermaphrodite and male). One of the most striking examples is the enormously, even 15-fold, reduced duration of mating in *C. elegans* when compared to its obligatory outcrossing relative *C. remanei* and, associated with it, similarly reduced rate of successful fertilization (*Chasnov & Chow, 2002*; *Chasnov, 2013*; *Garcia, LeBoeuf & Koo, 2007*). The mating attempts are short and inefficient at least partly because hermaphrodites are not susceptible to the soporific factor, which in dioecious *Caenorhabditis* species causes immobilization of females during sexual encounters (*Garcia, LeBoeuf & Koo, 2007*). Additionally, *C. elegans* hermaphrodites can easily escape from male copulation attempts or even eject male sperm if inseminated (*Kleemann & Basolo, 2007*). Overall, these and other related traits (cf. *Chasnov & Chow, 2002*; *Cutter, Morran & Phillips, 2019*) render outcrossing highly inefficient in *C. elegans*. Hallmarks of selfing syndrome can also be found at the genomic level. The estimated genome sizes of selfing *Caenorhabditis* species (*C. elegans*, 100.4 Mb; *C. briggsae,* 108 Mb; *C. tropicalis,* 79 Mb) are 12–40% smaller than these of their outcrossing relatives (*C. remanei*, 131 Mb; *C. brenneri*, 135 Mb; *C. japonica*, 135 Mb; *C. inopinata,* 123 Mb; *C. nigoni,* 129 Mb) (*Fierst et al., 2015*; *Kanzaki et al., 2018*; *Yin et al., 2018*). Transcriptomes of *C. elegans* and *C. briggsae* are substantially smaller than in *C. remanei, C. brenneri, C. japonica* and *C. nigoni*, with genes associated with sex-biased expression in outcrossing species being particularly likely to be missing in the selfers (*Thomas, Woodruff & Haag, 2012*; *Yin et al., 2018*). Thus, both complexity and sexual specialization of genomes and gene expression appear to have decreased in selfing *Caenorhabditis* lineages.

To see if the degenerated reproductive traits can re-evolve, reversing selfing syndrome, obligatory outcrossing can be re-introduced to *C. elegans* populations. This is achieved by blocking sperm production in hermaphrodites, by introgressing a homozygous loss of function mutation in one of the genes in hermaphrodite sperm development pathway, *e.g., fog-2*. This way, hermaphrodites become functional females which can only reproduce *via* outcrossing with males. Due to the XX/X0 sex determination system in *C. elegans*, outcrossing results in ∼1:1 male:female progeny. Thus, the proportion of males in obligatorily outcrossing population is increased to ∼50% (*Anderson, Morran & Phillips, 2010*; *Schedl & Kimble, 1988*). Such alteration of the mating system may shed light on how traits connected with the reproductive system evolve under laboratory conditions and whether they can be restored in species with selfing syndrome.

The experiment described in this article was part of a larger-scale research program (*Antoł et al., 2022a*; *Antoł et al., 2022b*; *Antoł et al., 2023*), in which we carried out experimental evolution with both wild-type (androdioecious) and *fog-2* (obligatorily

outcrossing) populations, starting from ancestors nearly devoid of genetic variation (isogenic), derived from *C. elegans* strain N2, which had been used in research for many decades and had undergone a long-term laboratory adaptation (*Sterken et al., 2015*). The main goals of the program were to study (i) how the reproductive system affects adaptation to a stressful novel environmental condition (increased ambient temperature) and (ii) how reproductive traits affected by selfing syndrome evolve after reversal to outcrossing. We chose the N2 strain in the hope that this would prevent confounding effects of adaptation to laboratory conditions—which is sometimes a problem in experimental evolution studies (*Teotónio et al., 2017*) (albeit this hope later proved to be unfulfilled, *Antoł et al., 2022a*). We further chose to use isogenic starting populations for two reasons. The primary one is not relevant to this particular article (cf. below) but was important for the broader goals of our research program: namely, to minimize differences in genetic background between wild type *vs.* obligatorily outcrossing ancestral populations (should genetically variable starting strain be used, such differences would inevitably arise, through segregation, during the process of deriving the obligatorily outcrossing populations). Secondly, low levels of standing genetic variation are generally characteristic of *C. elegans*, due to its primarily selfing mode of reproduction which enables (nearly) clonal expansions of single genotypes and associated genome-wide selective sweeps (*Andersen et al., 2012*). For this reason, experimental evolution studies featuring starting populations with high genetic diversity had to rely on constructing such populations by crossing several divergent isolates (*e.g.*, *Palopoli et al., 2015*; *Teotonio et al., 2012*). Because in our study the initial genetic variation was very low, the emergence of new adaptations would only depend on new mutations. Thanks to that, this experiment has a great comparative value towards the studies in which ancestral populations with increased standing genetic variation were used.

Here, we focus specifically on the evolution of fertilization efficiency in the *fog-2* (obligatorily outcrossing) populations, evolving in either (1) optimal temperature (20 °C) or (2) stressfully elevated temperature (24 °C). In the experiment described below, we compared the ancestral populations, which had their reproductive system changed but did not go through experimental evolution, with populations that evolved for over 100 generations in the new reproductive system. As outlined above, fertilizations in *C. elegans* are highly problematic. Under obligatory outcrossing, however, successful copulations are necessary for reproduction. Thus, we expected that adaptation to this reproductive system would lead, over generations, to an increase in fertilization efficiency.

As mentioned previously, our experimental evolution started from isogenic ancestors, any evolutionary change would be dependent on *de novo* mutations, occurring in each evolving population independently. Therefore, in our analyses, we were specifically interested assessing divergence from ancestors, with respect to fertilization efficiency, at the level of individual evolving populations, rather than averaging over them. Pinpointing specifically which (if any) populations are displaying evolutionary change would provide a base for subsequent more detailed investigations into underlying phenotypic and genetic mechanisms. However, investigating the level of individual populations is also necessarily associated with performing multiple comparisons (assessing divergence from ancestor separately for each evolving population), thus raising the risk of obtaining false positive

results. More generally, we believe that within-study reproducibility assessment is critical in the face of what is most commonly known as "replication crisis" in science (*Baker, 2016*; *Branch, 2019*; *Errington et al., 2021*; *Goodman, Fanelli & Ioannidis, 2016*; *Ioannidis, 2005*; *Moonesinghe, Khoury & Janssens, 2007*; *Parker, 2013*, cf. Discussion). Therefore, we assayed each evolving population, along with its ancestor, in 2–3 independent blocks, in order to assess both the reproducibility of our estimates and—as the other side of the same coin—their variability among blocks.

## MATERIALS & METHODS

### Strains and experimental evolution

We used the common laboratory-adapted *C. elegans* strain N2 (*Sterken et al., 2015*), obtained from the Caenorhabditis Genetics Center (CGC). From this strain, we derived replicate isogenic lines by 20 generations of single hermaphrodite transfers. As mentioned above, while the overall scope of our experimental evolution project was broader, including obligatorily outcrossing (*fog-2*) populations as well as those with wild type reproductive system, only the former were included in the fertilization experiment described in this article. Thus, the procedures described below refer only to the *fog-2* populations.

To create obligatorily outcrossing ancestral populations for experimental evolution, we introgressed fog-2(q71) mutation from strain JK574 independently into three of the abovementioned isolines (henceforth called isolines 6, 8 and 9). The introgression procedure followed *Teotonio et al. (2012)*, for more details see also *Plesnar-Bielak et al. (2017)*.

Each ancestral population was allowed to expand before being split into multiple sub-samples, some of which were banked at −80 °C, while the others were assigned to environmental treatments used for the experimental evolution (EE).

For the experimental evolution (EE), we applied two environmental treatments: 20 °C (standard laboratory temperature for *C. elegans* maintenance) and 24 °C (stressfully elevated temperature). Evolving populations were cultured in 14 cm ø Petri dishes with standard nematode growth medium (NGM) seeded with standard *E. coli* strain OP50 (*Brenner, 1974*), and transferred onto fresh plates every generation, with population size kept at *ca.* 10,000 individuals. To do this, transfers were performed using filters with 15 µm eyelets, which only let small larvae (L1-L2) through. Animals were washed from plates with 4 ml of S basal solution (*Stiernagle, 2006*) and the liquid with animals was placed on a filter positioned on 50 ml falcon. The filtered liquid containing L1-L2 larvae was vortexed (to achieve their even distribution) and the number of animals was counted in 2–3 drops of 1 µl each. Based on this count, the volume of liquid containing 10,000 individuals was estimated, and placed on a fresh plate seeded with bacteria. Transfers were made every *ca.* 3 days in populations kept in 24 °C and every *ca.* four days in populations kept in 20 °C, which referred to one generation cycle. Every *ca.* 12 generations, samples of the evolving populations (distributed into five separate vials per population) were frozen and kept in −80 °C for further assays (*Antoł et al., 2022a*). This procedure also prevented the loss of EE populations which would otherwise be lost due to cross-contamination, reversal of

outcrossing populations to selfing driven by gene conversion (*Katju et al., 2008*; *Antoł et al., 2022b*), or chance events. In such cases, a population was re-started from samples banked at an earlier time point (cf. *Antoł et al., 2022b*). Each population was evolving for at least 100 generations before being assayed in the experiments described below.

## Fertilization performance assay

Altogether, seven populations evolving in 20 °C and 12 populations evolving in 24 °C were included in the fertilization assay (Table 1), along with the three ancestral populations. The assay was performed using animals obtained from frozen samples of the evolved and ancestral populations described above (strains and experimental evolution section). In order to assess the replicability of our results, each evolved population was assayed in 2–3 independent blocks (we aimed for three; however, in several cases a population was lost from a block due to technical problems such as thawing failure or contamination), each time alongside its ancestral population (cf. Data analysis). Due to the amount of work involved, it was not possible to assay more than several populations in a single block. Thus, altogether the assay was performed in 17 replicate blocks, although the first block was excluded from the analysis due to technical problems. Each block contained 2–4 evolved populations from the same isoline along with their ancestral population. The genetic background (isoline) of each evolved population, along with its temperature treatment, blocks it was assayed in, and generation number are listed in Table 1. In each block, both evolved and ancestral populations were thawed from new vials, to make sure they went through the same number of generation transfers in each assay repetition. Unfortunately, in some cases, the evolved populations could not be obtained from the same freezing (generation) in all blocks because we had run out of stock for this particular time point. In these cases, the animals were thawed from other generations (see Table 1).

The preparatory stages of the assay are shown in Fig. 1A.

To prepare the populations for the assay, one frozen vial (per population) containing animals was thawed, placed on a Petri dish and incubated at 20 °C overnight. Because the survival rate during freezing can be low, the number of animals on each plate was checked on the following day, to make sure the initial population size was bigger than 100 individuals. If the number of animals was smaller, an additional vial with animals was placed on the same dish in order to keep the initial number of animals above 100 individuals. After this, the populations were left for five days at 20 °C to recover from freezing and start reproducing. After the recovery period and before the onset of the assay, defrosted populations went through three generations of transfers in order to minimalize the effects of freezing on the assayed phenotypes. The transfers are presented in Fig. 1A as days 6, 8 and 11. First, 6 days after thawing, each population was transferred onto a new plate using the chunk method (*Lewis & Fleming, 1995*), *i.e.,* by cutting a piece of agar containing animals from the original dish and placing it on a freshly prepared one. At this point, each evolved population was placed into the temperature of its prior evolution (20 °C or 24 °C). In blocks including evolved populations from only one temperature treatment, thawed sample of ancestral population was placed in the same temperature as them, whereas in blocks which included populations evolving in two different temperatures, the thawed

**Table 1 Evolving populations used in the experiment, along with their temperature of evolution, source isoline, numbers of blocks the were assayed in, and generation of evolution.** Cases, when a population was thawed from a different generation are marked with an underscore marking the block number.

| Temperature | Isoline | Population | Block | Generation |
|---|---|---|---|---|
| 20 | Iz8 | K02 | 3 | 127 |
| | | | 12 | 127 |
| | | | 16 | 127 |
| | | K12 | 3 | 112 |
| | | | 12 | 112 |
| | | | 16 | 112 |
| | | K25 | 4 | 141 |
| | | | 12 | 141 |
| | | | 16 | 141 |
| | | K54 | 4 | 128 |
| | | | 12 | 128 |
| | | | 16 | 128 |
| | Iz6 | K60 | 6 | 165 |
| | | | 11 | 165 |
| | | K28 | 6 | 165 |
| | | | 11 | 165 |
| | Iz9 | K29 | 2 | 165 |
| | | | 11 | 165 |
| | | | 17 | 165 |
| | | E01 | 6 | 113 |
| | | | 9 | 113 |
| | | E02 | 6 | 143 |
| | | | 9 | 143 |
| | | | 14 | 158 |
| | | E03 | 7 | 112 |
| | | | 9 | 112 |
| | | | 15 | 141 |
| | | E05 | 7 | 143 |
| | | | 9 | 143 |
| | | | 17 | 143 |
| | | E06 | 5 | 143 |
| | | | 14 | 131 |
| | | | 15 | 131 |
| | | E08 | 7 | 143 |
| | | | 8 | 143 |
| | | | 17 | 143 |
| | | E09 | 5 | 143 |
| | | | 8 | 143 |
| | | | 15 | 143 |

**Table 1** (*continued*)

| Temperature | Isoline | Population | Block | Generation |
|---|---|---|---|---|
| | | | 5 | 143 |
| | | E12 | 15 | 164 |
| | | | 17 | 164 |
| | | | 3 | 116 |
| | | E14 | 10 | 116 |
| | | | 13 | 116 |
| | Iz8 | E17 | 4 | 144 |
| | | | 10 | 144 |
| | | | 4 | 143 |
| | | E18 | 10 | 143 |
| | | | 13 | 158 |
| | Iz9 | E34 | 2 | 112 |
| | | | 13 | 112 |

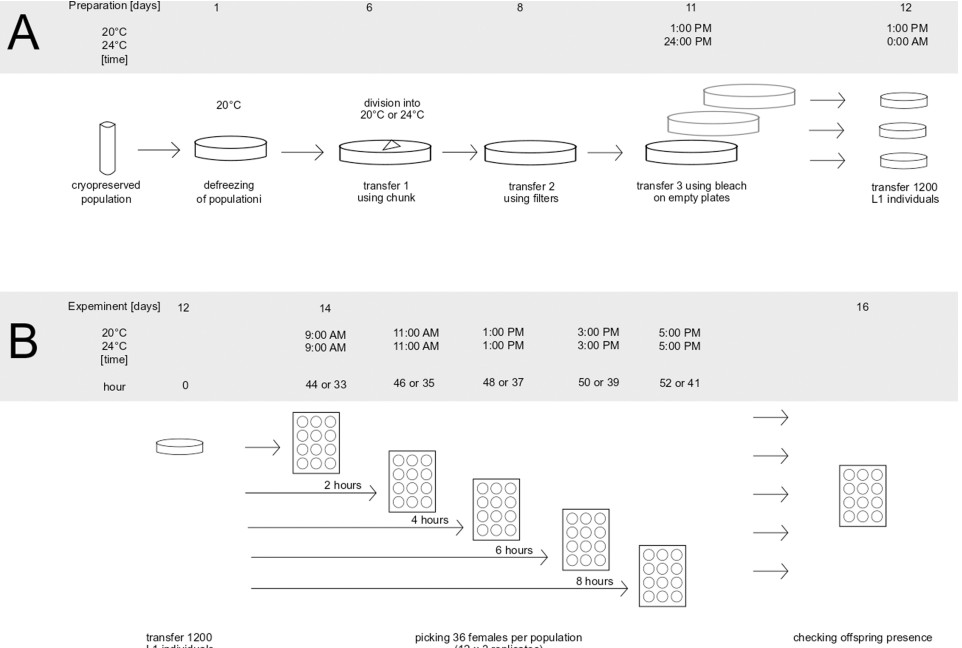

**Figure 1** Preparation of population for the experiment (A) and graphical representation of experiment (B).

sample of ancestral population was divided in two and placed into both temperatures. The second transfer (at day 8 after thawing) was performed using the filter method, the same as was used during experimental evolution (see above). Similarly, 10,000 larvae from each population were placed on a fresh dish. The last transfer (day 11) was performed using the bleach method (*Stiernagle, 2006*): treating animals with hypochlorite solution which kills and dissolves all adults and larvae, leaving only eggs (protected by shells) intact. Eggs after bleach were placed on empty Petri dishes, which causes newly hatched L1 larvae to go into
**Table 2  The results of data analysis for two models made in blocks for each population separately.**

| Temp (°C) | Isoline | Population | Block | Slope.ev | Slope.anc | Slope.diff | Slope.ratio | Slope. $p$ | Fisher slope. $p$ | Mean.ev | Mean.anc | Mean.diff | Mean.ratio | Mean. $p$ | Fisher mean. $p$ |
|---|---|---|---|---|---|---|---|---|---|---|---|---|---|---|---|
|  |  |  | 3 | 0.082 | 0.065 | 0.017 | 1.261 | 0.439 |  | 0.190 | 0.175 | 0.015 | 1.089 | 0.877 |  |
|  |  | K02 | 12 | 0.106 | 0.061 | 0.045 | 1.735 | 0.102 |  | 0.541 | 0.254 | 0.288 | 2.133 | 0.038* |  |
|  |  |  | 16 | 0.040 | 0.052 | −0.012 | 0.772 | 0.474 |  | 0.100 | 0.124 | −0.024 | 0.808 | 0.725 |  |
|  |  |  | 4 | 0.111 | 0.112 | −0.002 | 0.987 | 0.930 |  | 0.367 | 0.356 | 0.012 | 1.033 | 0.929 |  |
|  |  | K12 | 12 | 0.106 | 0.061 | 0.044 | 1.722 | 0.180 |  | 0.339 | 0.254 | 0.086 | 1.337 | 0.588 |  |
|  |  |  | 16 | 0.069 | 0.052 | 0.017 | 1.332 | 0.274 |  | 0.217 | 0.124 | 0.093 | 1.751 | 0.244 |  |
|  | Iz8 |  | 4 | 0.110 | 0.112 | −0.003 | 0.975 | 0.836 |  | 0.450 | 0.356 | 0.094 | 1.266 | 0.453 |  |
|  |  | K25 | 12 | 0.097 | 0.061 | 0.036 | 1.583 | 0.223 |  | 0.299 | 0.254 | 0.045 | 1.179 | 0.734 |  |
|  |  |  | 16 | 0.039 | 0.052 | −0.013 | 0.755 | 0.461 |  | 0.099 | 0.124 | −0.025 | 0.800 | 0.718 |  |
|  |  |  | **4** | **0.118** | **0.112** | **0.006** | **1.049** | **0.731** |  | **0.500** | **0.356** | **0.144** | **1.406** | **0.279** |  |
| 20 |  | **K54** | **12** | **0.109** | **0.061** | **0.048** | **1.778** | **0.051** | **0.193** | **0.332** | **0.254** | **0.078** | **1.308** | **0.548** | **0.580** |
|  |  |  | **16** | **0.070** | **0.052** | **0.017** | **1.334** | **0.350** |  | **0.165** | **0.124** | **0.042** | **1.336** | **0.619** |  |
|  |  | K60 | 6 | 0.099 | 0.106 | −0.007 | 0.933 | 0.753 |  | 0.324 | 0.322 | 0.002 | 1.005 | 0.99 |  |
|  |  |  | 11 | 0.106 | 0.031 | 0.075 | 3.408 | 0.001* |  | 0.292 | 0.073 | 0.219 | 4.000 | 0.038 |  |
|  | Iz6 |  | **6** | **0.112** | **0.106** | **0.007** | **1.065** | **0.651** | **0.374** | **0.404** | **0.322** | **0.082** | **1.255** | **0.511** | **0.341** |
|  |  | **K28** | **11** | **0.052** | **0.031** | **0.021** | **1.661** | **0.184** |  | **0.151** | **0.073** | **0.078** | **2.068** | **0.205** |  |
|  |  |  | 2 | 0.053 | 0.018 | 0.035 | 2.937 | 0.060 |  | 0.156 | 0.05 | 0.106 | 3.121 | 0.110 |  |
|  | Iz9 | K29 | 11 | 0.111 | 0.029 | 0.082 | 3.810 | 0.001* |  | 0.383 | 0.450 | −0.067 | 0.852 | 0.682 |  |
|  |  |  | 17 | 0.040 | 0.033 | 0.007 | 1.224 | 0.562 |  | 0.101 | 0.088 | 0.012 | 1.139 | 0.816 |  |

**Table 2** (*continued*)

| Temp (°C) | Isoline | Population | Block | Slope.ev | Slope.anc | Slope.diff | Slope.ratio | Slope. *p* | Fisher slope. *p* | Mean.ev | Mean.anc | Mean.diff | Mean.ratio | Mean. *p* | Fisher mean. *p* |
|---|---|---|---|---|---|---|---|---|---|---|---|---|---|---|---|
| | | **E01** | 6 | **0.047** | **0.028** | **0.019** | **1.674** | **0.091** | 0.017* | **0.162** | **0.076** | **0.087** | **2.146** | **0.099** | 0.074 |
| | | | 9 | **0.024** | **0.009** | **0.015** | **2.750** | **0.026*** | | **0.056** | **0.017** | **0.039** | **3.265** | **0.141** | |
| | | **E02** | 6 | **0.058** | **0.028** | **0.030** | **2.052** | **0.030*** | | **0.154** | **0.076** | **0.079** | **2.043** | **0.198** | |
| | | | 9 | **0.039** | **0.009** | **0.031** | **4.594** | **0.040*** | 0.010* | **0.106** | **0.017** | **0.088** | **6.147** | **0.085** | 0.000* |
| | | | 14 | **0.025** | **0.008** | **0.017** | **3.000** | **0.179** | | **0.222** | **0.028** | **0.194** | **8.000** | **0.000*** | |
| | | **E03** | 7 | **0.043** | **0.013** | **0.030** | **3.320** | **0.004*** | | **0.117** | **0.034** | **0.083** | **3.412** | **0.064** | |
| | | | 9 | **0.022** | **0.009** | **0.013** | **2.515** | **0.047*** | 0.006* | **0.049** | **0.017** | **0.032** | **2.868** | **0.200** | 0.130 |
| | | | 15 | **0.051** | **0.043** | **0.008** | **1.194** | **0.594** | | **0.133** | **0.094** | **0.039** | **1.412** | **0.560** | |
| | | **E05** | 7 | **0.034** | **0.013** | **0.021** | **2.643** | **0.011*** | 0.014* | **0.083** | **0.034** | **0.048** | **2.406** | **0.179** | 0.194 |
| | | | 9 | **0.018** | **0.009** | **0.010** | **2.132** | **0.182** | | **0.045** | **0.017** | **0.028** | **2.618** | **0.269** | |
| 24 | Iz6 | E06 | 5 | 0.043 | 0.014 | 0.029 | 3.090 | 0.008* | | 0.138 | 0.042 | 0.096 | 3.302 | 0.036* | |
| | | | 14 | 0.008 | 0.008 | 0.000 | 1.000 | 1.000 | | 0.017 | 0.028 | −0.011 | 0.600 | 0.426 | |
| | | | 15 | 0.015 | 0.043 | −0.028 | 0.355 | *0.027* | | 0.039 | 0.094 | −0.056 | 0.412 | 0.249 | |
| | | **E08** | 7 | **0.031** | **0.013** | **0.018** | **2.417** | **0.096** | 0.052 | **0.080** | **0.034** | **0.046** | **2.338** | **0.246** | 0.203 |
| | | | 8 | **0.052** | **0.021** | **0.031** | **2.496** | **0.096** | | **0.134** | **0.050** | **0.084** | **2.671** | **0.208** | |
| | | E09 | 5 | 0.028 | 0.014 | 0.014 | 2.019 | 0.087 | | 0.097 | 0.042 | 0.055 | 2.324 | 0.094 | |
| | | | 8 | 0.023 | 0.021 | 0.002 | 1.107 | 0.834 | | 0.081 | 0.050 | 0.031 | 1.622 | 0.407 | |
| | | | 15 | 0.021 | 0.043 | −0.022 | 0.484 | 0.099 | | 0.050 | 0.094 | −0.044 | 0.529 | 0.387 | |
| | Iz6 | E12 | 5 | 0.060 | 0.014 | 0.046 | 4.314 | 0.002* | | 0.176 | 0.042 | 0.134 | 4.206 | 0.031* | |
| | | | 15 | 0.042 | 0.043 | −0.001 | 0.968 | 0.921 | | 0.117 | 0.094 | 0.022 | 1.235 | 0.709 | |
| | | **E14** | 3 | **0.047** | **0.034** | **0.013** | **1.390** | **0.254** | | **0.138** | **0.096** | **0.042** | **1.434** | **0.442** | |
| | | | 10 | **0.049** | **0.003** | **0.047** | **17.800** | **0.000*** | 0.000* | **0.305** | **0.006** | **0.300** | **54.982** | **0.000*** | 0.000* |
| | | | 13 | **0.032** | **0.008** | **0.024** | **3.841** | **0.025*** | | **0.068** | **0.022** | **0.046** | **3.068** | **0.225** | |
| | Iz8 | E17 | 4 | 0.022 | 0.074 | −0.051 | 0.303 | *0.003* | | 0.055 | 0.300 | −0.245 | 0.183 | *0.002* | |
| 24 | | | 10 | 0.014 | 0.003 | 0.011 | 5.000 | 0.054 | | 0.028 | 0.006 | 0.022 | 5.000 | 0.238 | |
| | | E18 | 4 | 0.060 | 0.074 | −0.014 | 0.816 | 0.534 | | 0.218 | 0.300 | −0.082 | 0.727 | 0.385 | |
| | | | 10 | 0.080 | 0.003 | 0.078 | 28.955 | 0.000* | | 0.433 | 0.006 | 0.427 | 77.909 | 0.000* | |
| | | | 13 | 0.024 | 0.008 | 0.016 | 2.879 | 0.199 | | 0.203 | 0.022 | 0.180 | 9.114 | 0.000* | |
| | Iz9 | E34 | 2 | 0.080 | 0.028 | 0.052 | 2.886 | 0.000* | | 0.289 | 0.061 | 0.228 | 4.736 | 0.003* | |
| | | | 13 | 0.015 | 0.021 | −0.006 | 0.733 | 0.649 | | 0.033 | 0.042 | −0.008 | 0.800 | 0.832 | |

**Notes.**

Slope.ev, slope estimate for a given evolved population in a given block; slope.anc, slope estimate for the ancestral population in the same block; slope.diff, difference between the former and the latter; slope.ratio, ratio of the former to the latter (analogously for means); slope. *p*, *P* value for the interaction term in model 1; mean. *p*, *P* value for the interaction term in model 2.

Statistically significant values ($P < 0.05$) are marked with: asterisk for positive coefficient estimates or italic font for negative coefficient estimates. Underscore marking block number means that in that block, the population it applies to was thawed from a different generation than in the previous block(s) (cf. Table 1 for details). Bolded fonts are marking populations that in all blocks had positive slope and mean differences (slope.diff & mean.diff). For those populations, the *P* values (for means and slopes separately) were combined using the Fisher's method. The resulting combined *P* values are represented in columns Fisher slope. *p* and Fisher mean. *p* for slope. *p* and mean. *p* accordingly.

Palka et al. (2023), *PeerJ*, DOI 10.7717/peerj.15825

larval arrest until they are placed on food—which enabled synchronizing the animals right before the onset of the assay. Also at this stage, eggs from each population were evenly split onto three separate plates, creating three replicates per population for the subsequent assay. After 24 h (day 12), the L1 larvae were placed on new dishes with food. To do this, the animals were washed from each plate, the resulting suspension was vortexed to achieve uniform distribution of larvae in the liquid, and the number of animals in the liquid from each dish was scored independently in three 1 μl drops. Based on the scored numbers of individuals in drops, the amount of suspension containing an estimated 1,200 individuals was seeded on a six cm ø Petri dish containing food. This number corresponds to the density of animals during experimental evolution (10,000 individuals per 14 ø cm plate). Populations were seeded in 10 min intervals (*e.g.*, 1:00 PM–3 replicates of population K12, 1:10 PM 3 replicates of population K02 *etc.*). The order in which the populations and replicates within the populations were seeded was noted and followed the next day when isolating nematodes. This was done in order to minimize differences between populations (and replicates within populations) in the length of time spent on population plates.

The dishes were coded, and animals were left to grow in their corresponding temperature for 44 h at 20 °C and for 33 h at 24 °C. As established *via* pilot observations, these intervals corresponded to the time needed for the majority of the animals to reach the young adult stage (with sporadic L4 larvae still present) in the respective temperatures.

The assay scheme is depicted on Fig. 1B. The assay started 44 h (at 20 °C) or 33 h (at 24 °C) after L1 seeding, at a moment when the majority of animals were young adults and some L4 larvae were still observed (Fig. 1B: hour since adulthood "0"). At this point, 12 females per replicate plate (12 × 3 replicates = 36 per population) were transferred into 12-well plates (one animal per well) by hand using a picker. The order of transfer was fixed the same way the populations were seeded. The same procedure was repeated after 2, 4, 6, and 8 h—each time transferring new females from the replicate plates into new wells. After two days the females were checked for offspring presence, indicating that a female had achieved fertilization prior to being isolated. Based on this, we calculated the fraction of fertilized females, out of all 12 (or occasionally fewer in rare cases when some females were lost in the process) isolated at a given hour from a given replicate. These fractions (henceforth termed: 'inseminated fraction') constituted a dependent variable in the following statistical analyses (see below).

## Data analysis

The obtained data were analysed using R studio (*RStudio Team, 2020*), using tidyverse (*Wickham et al., 2019*) and dplyr (*Wickham et al., 2020*) packages for data management and lm function for creating statistical models. The dependent variable in the analyses was 'inseminated fraction' (*i.e.,* fraction of females which turned out to be inseminated, out of all isolated from a given replicate at a given timepoint), which, in every assay block, was calculated for each of the 4 timepoints (cf. Fig. 1B) for each of the 3 replicates within each population assayed. Additionally, some of *P* values were compared with Fisher's method using poolr package (*Cinar & Viechtbauer, 2022*).

As outlined in the Introduction, we were specifically interested in (i) comparing individual evolved populations with their ancestors and (ii) assessing the reproducibility *vs.* variability of our estimates across replicate blocks. Thus, we performed multiple analyses, comparing each evolved population with its ancestral one separately for each block they were both assayed in. For each combination of evolving population x block, we ran two complementary analyses:

1. Fertilization rate over time: Inseminated fraction c population ⋆ hour, where 'population' was a factor with two levels: ancestral (in the intercept) and evolved, and 'hour' was treated as a continuous variable. In these models we were specifically interested in the interaction term, which tested for the difference between the evolved and ancestral populations in the slope of increase in the fraction of inseminated females over time, within the 8-hour time window analyzed in our experiment. Raw data, along with regression slopes, illustrating these analyses are included in Supplement S1.

2. Mean fertilization rate: Inseminated fraction ∼ population. These models simply compared the overall fractions of females inseminated over the course of the 8-hour window, ignoring the time dimension. Put simply, these analyses were addressing the question: regardless of the rate of its increase, is the fraction of females which got inseminated (within the time window covered by the assay) higher in the evolved population relative to its ancestor? (see Supplement S2 for illustration).

Subsequently, we compared the results obtained across the blocks for each evolved population in turn. Specifically, for each evolved population in each block, we looked at the differences between its and its ancestor's (1) fertilization rate over time (obtained from model 1. as the difference in slopes) and (2) mean fertilization rate (obtained from model 2. as the difference in means). In order to evaluate the magnitude of these differences relative to the ancestral baseline (*e.g.*, a +0.05 difference in slope or mean represents a 6-fold upward divergence if the ancestral value was 0.01, but only a 50% upward divergence if the ancestral value was 0.1), in each case we also calculated the ratios of evolved-to-ancestral slope and mean. The ratios are visualized in Fig. 2 (slopes) and in Fig. 3 (means), while all results from the described analysis are presented in Table 2.

If better fertilization efficiency has indeed evolved in some of our EE populations, we expected that these populations would consistently have higher scores than their ancestors in both measures, in all blocks they were assayed in. For populations which matched these criteria, we used Fisher's technique (*Sokal & Rohlf, 1995*) to combine the *P* values obtained from the analyses of separate blocks, in order to assess the statistical significance of this measure of divergence.

As explained above, the experiment was performed in 17 replicate blocks, but the first block had to be excluded from the analyses due to technical failure; hence the block numbering from 2 to 17. Additionally, in the last block (nr 17), at 24 °C virtually no fertilizations were observed across the 8 h time window, neither in the ancestral nor in the three evolved populations assayed (with the exception of a single inseminated female in one of the evolved populations, at hour 8). Therefore, these data could not be analyzed and are not included in Table 2; however, they are displayed on Fig. 4 and in Supplement S1 on panels K, M and O.

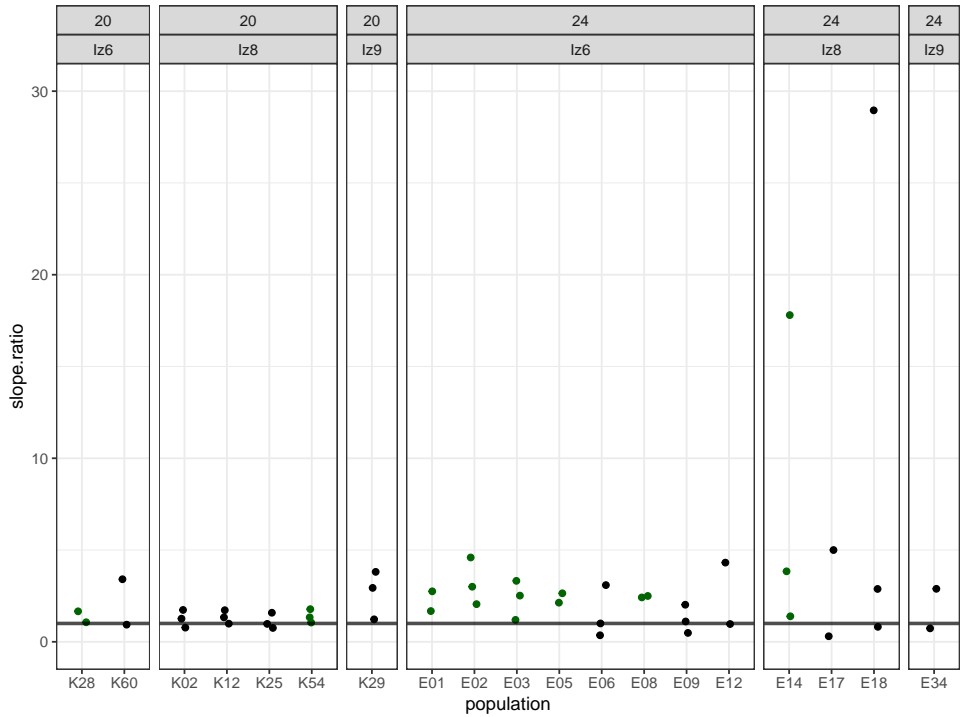

**Figure 2** **The ratios of evolved-to-ancestral slopes of fertilization rate over time.** Each data point represents ratio calculated for a given population (*x* axis) in one replicate block. Green colour marks populations that in all blocks had both slope and mean (cf. Fig. 3 & Table 2) scores higher than ancestors (ratios > 1).

## RESULTS

We investigated the evolution of fertilization efficiency in replicate *C. elegans* populations with genetically induced obligatory outcrossing. Following > 100 generations of experimental evolution at either optimal (20 °C) or elevated (24 °C) temperature, fertilization rates were assayed in the evolving populations, as well as in their ancestors who had the obligatory outcrossing introduced but did not go through experimental evolution. The assays tracked the fractions of fertilized females in age-synchronized populations, through 8 h since reaching adulthood. They were performed at the evolving populations' respective temperatures of evolution. In order to check the robustness of our measurements, each evolving population was assayed, along with its ancestor, in two or three independent replicate blocks; in each block, we compared its (i) slope of fertilization rate over time and (ii) mean fertilization rate to these of its ancestor, using linear models.

We identified eight populations in which slope and mean estimates were consistently higher than in their ancestors across all experimental blocks they were assayed in: six (out of the 12 assayed) evolving at 24 °C and two (out of seven assayed) evolving at evolving at 20 °C (Table 2, Figs. 2 and 3). As judged by the Fisher's method of pooling *P* values (*Sokal & Rohlf, 1995*), in two of the 24 °C populations (E02 and E14) these effects were statistically significant for both slopes and means, in three 24 °C populations (E01, E03 and

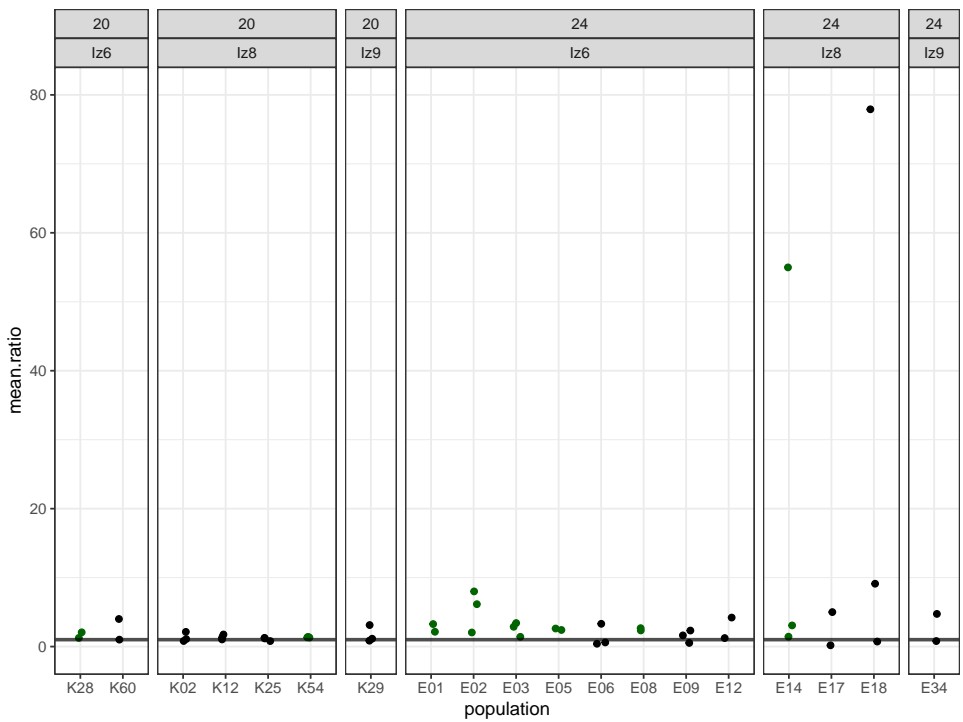

**Figure 3** **The ratios of evolved-to-ancestral mean fertilization rates.** Each data point represents ratio calculated for a given population (*x* axis) in one replicate block. Green colour marks populations that in all blocks had both mean and slope (cf. Fig. 2 & Table 2) scores higher than ancestors (ratios > 1).

E05) slope differences were statistically significant but mean differences were not, whereas in one 24 °C population (E08) and both 20 °C populations (K54 and K28) neither the slope nor mean differences were statistically significant (Table 2).

For the remaining populations, the effects recorded across blocks varied from positive (evolved population having higher scores than ancestor) to negative (evolved population having lower scores than ancestor). No evolving population showed consistently downwards divergence from ancestor across blocks (Table 2).

Furthermore, very clear in our data is high among-block variability in the populations' fertilization trajectories, (cf. Supplement S1) particularly well visible for ancestral populations (Fig. 4). Analogously, for the majority of the evolving populations, especially at 24 °C, the estimates of fertilization rate's divergence from ancestors also displayed substantial variability among blocks (Figs. 2 and 3, Table 2, see *e.g.*, populations E06, E14, E17, E18 and E34). Particularly illustrative of this variability are cases of populations E06 and E18 (Table 2, Supplement S1 L & S, Supplement S2 B, panels: E06 & E19). For E06, the ratios of evolving-to-ancestral slopes ranged from 0.35 in block 15 (*i.e.,* E06's slope of fertilization rate over time being 65% less steep than its ancestor's) to 3.09 in block 5 (E06's slope 3.09-fold steeper than the ancestor's), with the difference in both cases actually turning statistically significant. For E18, the ratios ranged from 0.82 (slopes) and 0.73 (means) in block 4 to 28.95 (slopes) and 77.91 (means) in block 10. The exceptionally

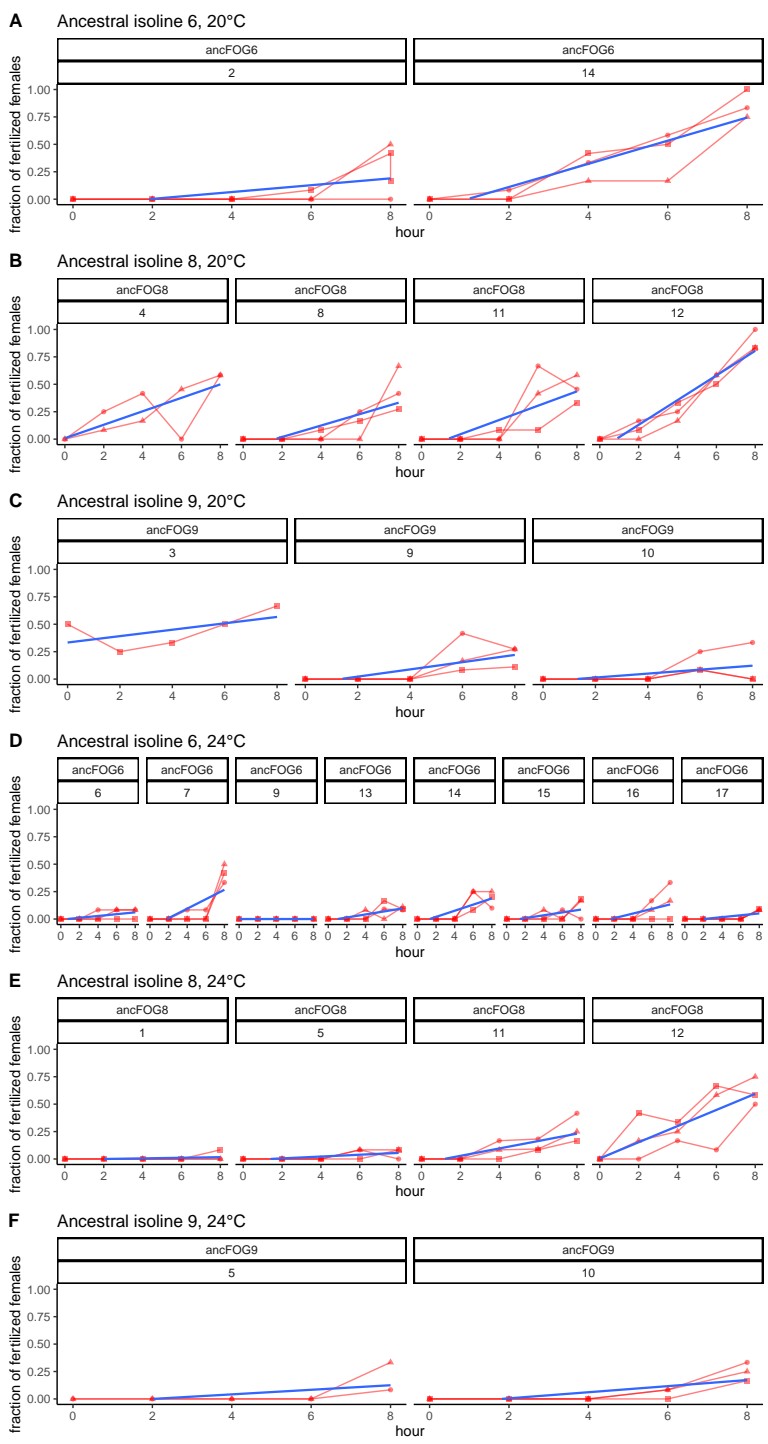

**Figure 4** **Fertilization trajectories of ancestral populations. Population (upper) and block (lower) IDs are presented in boxes at the top of the panels.** (A–C) Present results for 20 °C and D–F represent 24 °C. Plots are generated from raw data, blue lines represent slopes obtained from models 1 (cf. Methods: Data analysis for model description and Table 2 for coefficient estimates).

high ratios in block 10 were associated with the fact that in this block we observed almost no fertilizations in the ancestral population, except for a single inseminated female in one replicate at hour 8, while a number of inseminated females were found in E18 (as well as in the other two evolving populations assayed in this block—E14 and E17, for which high ratios were also consequently observed) (Supplement S1 panels P and R, Table 2).

Additionally, we made descriptive observations regarding the fertilization trajectories, revealing that the fertilization peaks in our populations were beyond our assay time-frame in both temperatures, but particularly so in 24 °C. First, we looked at which hour of the assay the first fertilizations were occurring. The number and percentage of replicates and populations in which first fertilizations were observed at consecutive timepoints are presented in Table 3. From this data, we see that at the beginning of experiment (hour "0") fertilizations were rare, occurring only in 9% of replicates (from 50% of the populations) at 20 °C and only in 6% of replicates (from 20% of populations) at 24 °C. Most of fertilizations events in 20 °C began during hours 2, 4 and 6 of experiment (in total 87.1% of replicates). This peak is shifted towards hours 4, 6 and 8 in 24 °C (in total 74.5% of replicates). There were also some replicates in which none of the females got inseminated through the 8 h of experiment. This is particularly visible in the elevated temperature, where in over 11% of replicates (from one third of populations) no successful fertilization was observed. This also occurred in one replicate from 20 °C. Secondly, within the 8th hour window starting at hour "0", only in few replicates, and only in 20 °C, the maximal (100%) fraction of inseminated females have been achieved. The difference between temperatures was substantial: on average, at 20°C, 60% of females were inseminated at the 8th hour of the assay, whereas at the 24 °C treatment, on average only 26% of females were inseminated by that time. To check if these fractions would increase over time, in the last three experimental blocks we additionally isolated females after 24 and/or 26 h since the onset of the assay (hour "0"/adulthood) and checked for offspring presence. Indeed, after this time, we observed an increment in the mean fertilization rate, which achieved 95% in the control temperature and 92% in the higher temperature. Results from the fraction of inseminated females after 24 h are presented in Fig. 5.

Data together with the code used in analyses are available in online repositories (Figshare—data and Zenodo—code).

## DISCUSSION

In *C. elegans*, evolutionary history of primarily selfing reproduction has rendered cross-fertilization inefficient relative to its obligatorily outcrossing relatives. We have predicted that *C. elegans* populations evolving under genetically induced obligatory outcrossing may, over generations, develop heightened cross-fertilization efficiency (contingent on the appearance of relevant genetic variants). In this study, we assayed fertilization rates of (i) 19 populations which had gone through >100 generations of evolution under obligatory outcrossing at either the standard laboratory temperature of 20 °C (seven populations) or elevated temperature of 24 °C (12 populations) and (ii) their ancestral populations (in which evolution was halted directly after the induction of obligatory outcrossing).

**Table 3 Numbers (#) and percentages (%) of replicates (reps) and populations (pops) in which first fertilized female(s) was/were observed at consecutive timepoints of the assay (first column).** Last verse shows cases where no fertilizations were observed throughout the 8-hour assay. In case of populations, the percentages sum up to >>100, that is because they were calculated separately for each timepoint, as percentage of populations in which the fertilization occurred in at least one replicate (*e.g.* hour "0" 5 populations where fertilization occurred/10 populations in total * 100% = 50%).

| First fertilized females observed at hour... | 20 °C | | | | 24 °C | | | |
|---|---|---|---|---|---|---|---|---|
| | #reps | %reps | #pops | %pops | #reps | %reps | #pops | %pops |
| 0 | 7 | 9.1 | 5 | 50.0 | 8 | 6.0 | 3 | 20.0 |
| 2 | 23 | 29.9 | 9 | 90.0 | 11 | 8.3 | 7 | 46.7 |
| 4 | 20 | 26.0 | 9 | 90.0 | 32 | 24.1 | 12 | 80.0 |
| 6 | 24 | 31.2 | 10 | 100.0 | 39 | 29.3 | 13 | 86.7 |
| 8 | 2 | 2.6 | 2 | 20.0 | 28 | 21.1 | 11 | 73.3 |
| None through the 8 h assay | 1 | 1.3 | 1 | 10.0 | 15 | 11.3 | 5 | 33.3 |

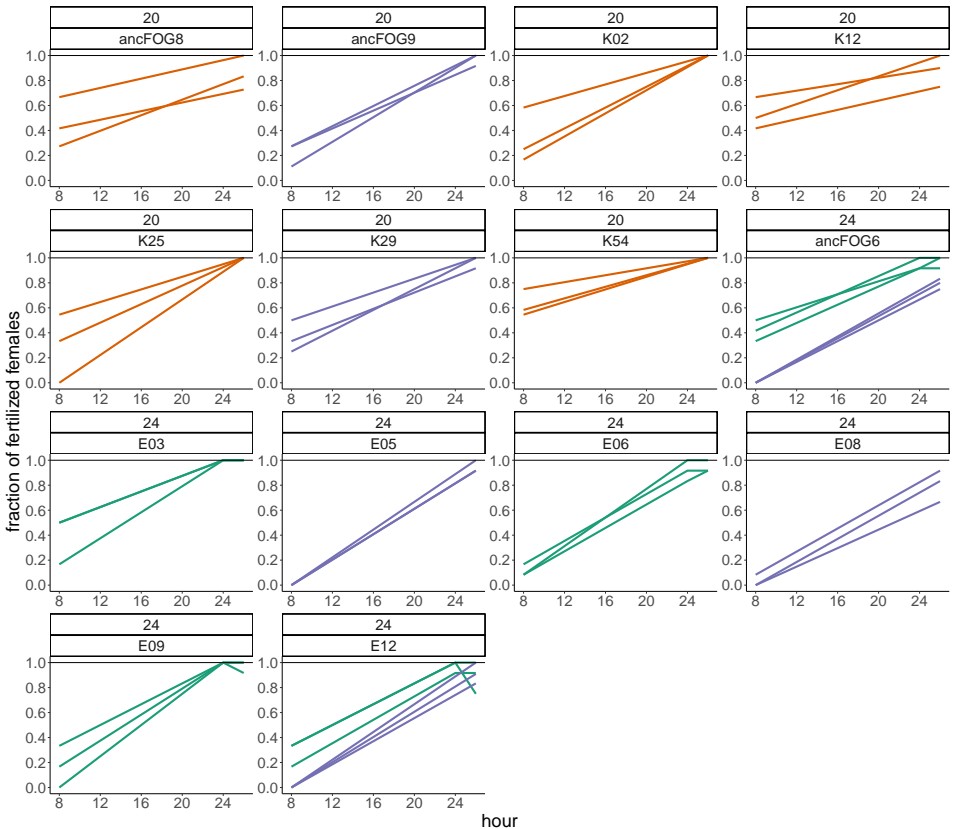

**Figure 5 Fractions of fertilized females recorded after 8 and 24–26 h from the beginning of the assay in blocks 15–17.** Different line colour represents block.

With replicated assays spanning 8 h since early adulthood in age-synchronized population samples, we estimated the divergence of the evolving populations from their ancestors using two measures of fertilization performance: the slope of fertilization rate over time and mean fertilization rate across the 8 h time window. Out of the 19 evolving populations assayed, we identified eight and six of them from 24 °C and 2 from 20 °C respectively, in which both measures of fertilization performance were consistently higher than in their ancestors across all replicate assay blocks they were scored in. In five of the 24 °C populations, these differences were statistically significant for either both slopes and means (populations E02 and E14) or slopes only (populations E01, E03 and E05). In one 24 °C population and both 20 °C populations, they were not statistically significant. Furthermore, in the remaining 11 populations (six from 24 °C and 5 from 20 °C), the effects recorded across blocks varied from positive (evolved population having higher fertilization rate measures than ancestor) to negative (evolved population having lower measures than ancestor) (cf. Table 2, Figs. 2 and 3). Based on this data, we conclude that 14 populations we assayed do not appear to have diverged from their ancestors in terms of fertilization efficiency, whereas five populations (E02, E14, E01, E03 and E05, all from 24 °C) have showed signatures of such divergence.

However, we want to be cautious with these conclusions due to the high among-block variability of divergence measures, which we observed in multiple evolving populations, particularly at the higher temperature (Table 2, Figs. 2 and 3). This high variability indicates that at this stage, the populations E02, E14, E01, E03 and E05 should be treated as candidate rather than showing conclusive evidence for having evolved increased fertilization rate. Furthermore, it also suggests that for some populations we might have not been able to detect increased fertilization rate which had in fact evolved. Thus, in order to robustly assess the evidence for our populations evolutionary responses—or lack thereof—we would need more assay replicates than the 2–3 featured in this study. Given the amount of work involved in the assays coupled with the high number of evolving populations, we were not able to have higher replication at this stage of the project. However, this may be achieved in the future starting with the smaller set of evolving populations including the five candidates we have identified.

High among-block variability in our data be related to uncontrolled variation in numerous micro-environmental factors affecting the nematodes' development and reproduction, operating both during preparatory stages (including the process of population freezing and thawing) and during the assay itself. For example, as we have observed repeatedly during our research, one of such factors is contamination—the presence of other bacteria or fungi, besides the worms' designated food source, on agar plates. Throughout our study, we were occasionally encountering problems with the contamination of agar plates, especially at 24 °C. Depending on contamination size and/or variant, it could restrict access to food and influence the time of animal development, thus affecting fertilization trajectories and consequently, divergence scores and their variability. An intriguing hypothesis which could be tested in further studies including obligatorily outcrossing *Caenorhabditis* species is that high variability of cross-fertilization dynamics may in itself be related to selfing syndrome, and that fertilization rates in

"true" outcrossers would be more robust to uncontrolled sources of variability. However, high among-block variability is a phenomenon we have observed also when assaying a trait unrelated to outcrossing—fitness of *C. elegans* populations with wild type (selfing) mode of reproduction (*Antoł et al., 2022a*). More generally, abundant biological variability belongs to the fundamental characteristics of all life. It also most certainly is an important contributor to the common failure to replicate research results in multiple scientific disciplines (cf., *e.g.*, *Hirschhorn et al., 2002*; *Lithgow, Driscoll & Phillips, 2017*; *Voelkl et al., 2020*).

Despite the caveats discussed above, we conclude (however cautiously) that based on data available at this stage, five populations evolving at 24 °C show patterns suggestive of increased fertilization rates relative to ancestors, whereas the majority of evolving populations (the remaining seven from 24 °C and all seven from 20 °C) do not. An important factor contributing to the lack of detected evolutionary response in most populations is the lack of genetic variation available for selection to act upon. In initially isogenic populations, the only source of genetic variation are randomly occurring *de novo* mutations. This limits evolutionary potential substantially, albeit by no means entirely: rapid evolutionary response attributed to new mutations have been reported by studies on various traits in various species, including, *e.g.*, fitness (*Denver et al., 2010*; *Antoł et al., 2022a*) and body size (*Azevedo et al., 2002*) in *C. elegans*, bristle number in *Drosophila melanogaster* (*Merchante, Caballero & López-Fanjul, 1995*) or song-related wing morphology in *Teleogryllus oceanicus* (*Pascoal et al., 2014*).

However, when studying outcrossing-related traits in *C. elegans*, as we did here, the shortage of relevant *de novo* variants may be aggravated by the fact that, as outlined in the Introduction, selfing syndrome in *C. elegans* is also manifested by genome shrinkage and, the loss of many genes with sexually specialized function (*Thomas, Woodruff & Haag, 2012*; *Yin et al., 2018*). Thus, some of the loci that historically regulated mating success may have been deleted from the genome, further decreasing the frequency of relevant *de novo* mutations by restricting the pool of genes in which they could appear. Presumably, more consistent response across populations may have been observed if genetically variable ancestors were used, as was the case in several earlier studies investigating experimental evolution of mating related traits in *C. elegans* (*LaMunyon & Ward, 2002*; *Palopoli et al., 2015*; *Teotonio et al., 2012*). Nevertheless, despite the lack of initial genetic diversity in this study, our data suggest that response to selection has occurred in the five candidate populations evolving in the higher temperature.

Moreover, we also noticed an interesting effect regarding the differences between the trajectories of fertilization at 20 °C *vs.* 24 °C. In general, during 8 h of the experiment, populations kept in 20 °C achieved higher fertilization rates than populations in the second treatment. Additional observations carried out in the last 3 assay blocks revealed that this difference declined on the next day (24 h–26 h since the timepoint designated as hour "0" in our study, marking the early adulthood of the majority of individuals in population). At this time, the fertilization rates were reaching over 90% in all cases, regardless of temperature. From previous studies, we know that in *C. elegans*, the elevated temperature is causing reduction in reproductive success in both wild-type and *fog-2*

mutants (*e.g.*, *Byerly, Cassada & Russell, 1976*; *Plesnar-Bielak et al., 2017*). In a previous study by our group, lifetime reproductive success was measured at optimal (20 °C) and elevated (25 °C) temperature, in pair matings (from *fog-2* populations) or individual hermaphroditic (from wild-type populations). Reduction in fitness caused by thermal stress was especially apparent in animals from *fog-2* populations, where a large fraction of pairs failed to produce offspring entirely (*Plesnar-Bielak et al., 2017*). The more prominent effect of high temperature in *fog-2* animals could arise *via* its effect on males, perhaps specifically on copulatory behaviours. Influence of high temperature on male fitness was also evident in study performed by *Petrella (2014)*, where she showed that the percentage of *C. elegans* males that produce progeny dropped to near zero when males were raised at 27 °C. Other study suggest that the 27 °C had effects on mating behaviour, sperm transfer and male tail morphology in males (*Nett, Sepulveda & Petrella, 2019*). The main difference between our and described studies concerned temperature, which in our case was lower (24 °C). Hence, in our study, the effect of elevated temperature could contribute to slower fertilization rates, although in a less drastic way than 27 °C or even 25 °C would. Another difference between studies concerned the number of animals which were used in the experiment. All described above experiments were done either on mating pairs or on a sample of several dozen animals. In our study, we decided to measure the fertilization rate by sampling females from populations with over 1,000 individuals. This means that even if the majority of males were failing at mating, most females could still be inseminated by those who were functional enough. However, the shift towards later fertilizations (as observed in our study), could perhaps be explained by more time being needed for fewer functional males to mate with the large number of females. Alternatively, another explanation for the observed differences in fertilization rates during the 8 h of the experiment could be developmental differences between temperatures.

Our fertilization assay began (hour ''0'') at the stage when vast majority of individuals in populations were young adults, with sporadic L4 individuals still present. This stage corresponded to 44 h after L1 larvae transfer at 20 °C and 33 h at 24 °C, as we established through pilot observations, based on visual differentiation between L4 & adult stages, which in *C. elegans* is precise. However, although we strived to be as precise as possible at pinpointing the developmentally identical stage for both temperatures, small differences in the proportions of adults *vs.* larvae could be neglected. Moreover, similarly to other phenotypes, developmental rate of individuals, as well as its variability among them, are not fixed within a temperature, but additionally affected by a number of other factors. As mentioned above, one of such factors is—the presence of other bacteria or/and fungi, besides the worms' designated food source, on agar plates. Throughout our study, we occasionally encountered problems with the contamination of agar plates. Such problems were occurring more frequently at 24 °C. Depending on contamination size and/or variant, it could restrict access to food and influence the time of animal development.

To summarize, our study has revealed considerable levels of variability in populations' fertilization trajectories. We have also identified 5 ''candidate'' populations which may have evolved increased fertilization rate relative to their ancestors. Such a small number of candidate populations could be due to a lack of initial genetic variation. This factor

combined with a relatively short duration of evolution ($\sim$100 generations) could contribute to observed low selection response. Further studies would be needed in order to either confirm or disprove these populations divergence from their ancestors and, potentially, investigate the underlying mechanisms. Other studies could also be designed to investigate the sources of the observed variation. For example, an assay with obligatory outcrossing species from the *Caenorhabditis* group could show if the variation is higher in *C. elegans* populations with altered reproductive type than in "true" outcrossers.

## CONCLUSIONS

We have predicted that through over 100 generations of obligatory outcrossing, populations may evolve heightened fertilization efficiency (contingent on the appearance of relevant genetic variants). Indeed, we have identified five populations in which such changes appear to have evolved. However, we want to be careful with this conclusion since our study has also revealed considerable levels of among-block variability in populations' fertilization trajectories, translating to analogous variability in the estimates of the evolving populations' divergence from ancestors. Thus, the populations we have identified should be treated as candidate, with more assay replications needed to either confirm or disprove their divergence and thus established whether further investigations into the underlying mechanisms may be warranted. At this stage, our primary insights concern the high levels of variability in our estimates, and the need for more careful and extensive treatment of biological variation in future studies (cf. *Voelkl et al., 2020*).

## ACKNOWLEDGEMENTS

We thank Wiesław Babik, Agata Plesnar-Bielak and our whole team for support and helpful comments during this work.

### Funding

This study was financed by the National Science Centre (Poland) grant 2017/26/E/NZ8/00879 to Zofia M. Prokop. The funders had no role in study design, data collection and analysis, decision to publish, or preparation of the manuscript.

### Grant Disclosures

The following grant information was disclosed by the authors:
National Science Centre (Poland): 2017/26/E/NZ8/00879.

### Competing Interests

The authors declare there are no competing interests.

### Author Contributions

- Joanna K. Palka conceived and designed the experiments, performed the experiments, analyzed the data, prepared figures and/or tables, authored or reviewed drafts of the article, and approved the final draft.

- Alicja Dyba performed the experiments, authored or reviewed drafts of the article, and approved the final draft.
- Julia Brzozowska performed the experiments, authored or reviewed drafts of the article, and approved the final draft.
- Weronika Antoł conceived and designed the experiments, performed the experiments, authored or reviewed drafts of the article, and approved the final draft.
- Karolina Sychta performed the experiments, authored or reviewed drafts of the article, and approved the final draft.
- Zofia M. Prokop conceived and designed the experiments, analyzed the data, prepared figures and/or tables, authored or reviewed drafts of the article, and approved the final draft.

### Data Availability

The raw data is available at Figshare: Palka, Joanna (2023). Dataset_to_Evolution_of_fertilization_ability_in_obligatorily_outcrossing_populations_of_Caenorhabditis_elegans.csv. figshare. Dataset. https://doi.org/10.6084/m9.figshare.22093832.v3.

The code for statistical analysis and plot generation is available at Zenodo: Joanna K. Palka, Alicja Dyba, Julia Brzozowska, Weronika Antoł, Karolina Sychta, & Zofia M. Prokop. (2023). Evolution of fertilization ability in obligatorily outcrossing populations of Caenorhabditis elegans. Zenodo. https://doi.org/10.5281/zenodo.7885219.

### Supplemental Information

Supplemental information for this article can be found online at http://dx.doi.org/10.7717/peerj.15825#supplemental-information.

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

# PeerJ

**Goodman SN, Fanelli D, Ioannidis JPA. 2016.** What does research reproducibility mean? *Science Translational Medicine* **8(341)**:341ps12–341ps12 DOI 10.1126/scitranslmed.aaf5027.

**Hirschhorn JN, Lohmueller K, Byrne E, Hirschhorn K. 2002.** A comprehensive review of genetic association studies. *Genetics in Medicine* **4(2)**:45–61 DOI 10.1097/00125817-200203000-00002.

**Ioannidis JPA. 2005.** Why most published research findings are false. *PLOS Medicine* **2(8)**:e124 DOI 10.1371/journal.pmed.0020124.

**Jarne P, Auld JR. 2006.** Animals mix it up too: the distribution of self-fertilization among hermaphroditic animals. *Evolution* **60(9)**:1816–1824 DOI 10.1554/06-246.1.

**Kanzaki N, Tsai IJ, Tanaka R, Hunt VL, Liu D, Tsuyama K, Maeda Y, Namai S, Kumagai R, Tracey A, Holroyd N, Doyle SR, Woodruff GC, Murase K, Kitazume H, Chai C, Akagi A, Panda O, Ke HM, Schroeder FC, Wang J, Berriman M, Sternberg PW, Sugimoto A, Kikuchi T. 2018.** Biology and genome of a newly discovered sibling species of *Caenorhabditis elegans*. *Nature Communications* **9(1)**:1–12 DOI 10.1038/s41467-018-05712-5.

**Katju V, Labeau EM, Lipinski KJ, Bergthorsson U. 2008.** Sex change by gene conversion in a *Caenorhabditis elegans* fog-2 Mutant. *Genetics* **672**:669–672 DOI 10.1534/genetics.108.090035.

**Kiontke K, Gavin NP, Raynes Y, Roehrig C, Piano F, Fitcht DHA. 2004.** *Caenorhabditis* phylogeny predicts convergence of hermaphroditism and extensive intron loss. *Proceedings of the National Academy of Sciences of the United States of America* **101(24)**:9003–9008 DOI 10.1073/pnas.0403094101.

**Kleemann GA, Basolo AL. 2007.** Facultative decrease in mating resistance in hermaphroditic *Caenorhabditis elegans* with self-sperm depletion. *Animal Behaviour* **74(5)**:1339–1347 DOI 10.1016/j.anbehav.2007.02.031.

**LaMunyon CW, Ward S. 2002.** Evolution of larger sperm in response to experimentally increased sperm competition in *Caenorhabditis elegans*. *Proceedings of the Royal Society B* **269(1496)**:1125–1128 DOI 10.1098/rspb.2002.1996.

**Lewis JA, Fleming JT. 1995.** Chapter 1: basic culture methods. *Methods in Cell Biology* **48(C)**:3–29 DOI 10.1016/S0091-679X(08)61381-3.

**Lithgow GJ, Driscoll M, Phillips P. 2017.** A long journey to reproducible results. *Nature* **548**:387–388 DOI 10.1038/548387a.

**Merchante M, Caballero A, López-Fanjul C. 1995.** Response to selection from new mutation and effective size of partially inbred populations. II. Experiments with *Drosophila melanogaster*. *Genetical Research* **66(3)**:227–240 DOI 10.1017/S0016672300034674.

**Moonesinghe R, Khoury MJ, Janssens ACJW. 2007.** Most published research findings are false—but a little replication goes a long way. *PLOS Medicine* **4(2)**:e28 DOI 10.1371/journal.pmed.0040028.

**Nett EM, Sepulveda NB, Petrella LN. 2019.** Defects in mating behavior and tail morphology are the primary cause of sterility in *Caenorhabditis elegans* males at high temperature. *Journal of Experimental Biology* **222(24)**:jeb208041 DOI 10.1242/jeb.208041.

**Palopoli MF, Peden C, Woo C, Akiha K, Ary M, Cruze L, Anderson JL, Phillips PC. 2015.** Natural and experimental evolution of sexual conflict within *Caenorhabditis* nematodes. *BMC Evolutionary Biology* **15(1)**:1–13 DOI 10.1186/s12862-015-0377-2.

**Parker TH. 2013.** What do we really know about the signalling role of plumage colour in blue tits? A case study of impediments to progress in evolutionary biology. *Biological Reviews* **88(3)**:511–536 DOI 10.1111/brv.12013.

**Pascoal S, Cezard T, Eik-Nes A, Gharbi K, Majewska J, Payne E, Ritchie MG, Zuk M, Bailey NW. 2014.** Rapid convergent evolution in wild crickets. *Current Biology* **24(12)**:1369–1374 DOI 10.1016/j.cub.2014.04.053.

**Petrella LN. 2014.** Natural variants of *C. elegans* demonstrate defects in both sperm function and oogenesis at elevated temperatures. *PLOS ONE* **9(11)**:e112377 DOI 10.1371/journal.pone.0112377.

**Plesnar-Bielak A, Labocha MK, Kosztyła P, Woch KR, Banot WM, Sychta K, Skarboń M, Prus MA, Prokop ZM. 2017.** Fitness effects of thermal stress differ between outcrossing and selfing populations in *Caenorhabditis elegans*. *Evolutionary Biology* **44(3)**:356–364 DOI 10.1007/s11692-017-9413-z.

**RStudio Team. 2020.** RStudio: integrated development environment for R. *Available at http://www.rstudio.com/.*

**Schedl T, Kimble J. 1988.** fog-2, a germ-line-specific sex determination gene required for hermaphrodite spermatogenesis in *Caenorhabditis elegans*. *Genetics* **119**:43–61 DOI 10.1093/genetics/119.1.43.

**Shimizu KK, Tsuchimatsu T. 2015.** Evolution of selfing: recurrent patterns in molecular adaptation. *Annual Review of Ecology, Evolution, and Systematics* **46(1)**:593–622 DOI 10.1146/annurev-ecolsys-112414-054249.

**Sokal RR, Rohlf FJ. 1995.** *Biometry: the principles and practice of statistics in biological research.* 3rd Edition. New York: W.H. Freeman and Co.

**Sterken MG, Snoek LB, Kammenga JE, Andersen EC. 2015.** The laboratory domestication of *Caenorhabditis elegans*. *Trends in Genetics* **31(5)**:224–231 DOI 10.1016/j.tig.2015.02.009.

**Stiernagle T. WormBook. 2006.** Maintenance of *C. elegans*. In: *The C. elegans Research Community, WormBook. Available at http://www.wormbook.org* DOI 10.1895/wormbook.1.101.1.

**Teotonio H, Carvalho S, Manoel D, Roque M, Chelo IM. 2012.** Evolution of outcrossing in experimental populations of *Caenorhabditis elegans*. *PLOS ONE* **7(4)**:e35811 DOI 10.1371/journal.pone.0035811.

**Teotónio H, Estes S, Phillips PC, Baer CF. 2017.** Experimental evolution with *Caenorhabditis* nematodes. *Genetics* **206(2)**:691–716 DOI 10.1534/genetics.115.186288.

**Thomas CG, Woodruff GC, Haag ES. 2012.** Causes and consequences of the evolution of reproductive mode in *Caenorhabditis* nematodes. *Trends in Genetics* **28(5)**:213–220 DOI 10.1016/j.tig.2012.02.007.

**Voelkl B, Altman NS, Forsman A, Forstmeier W, Gurevitch J, Jaric I, Karp NA, Kas MJ, Schielzeth H, Van de Casteele T, Würbel H. 2020.** Reproducibility of animal

research in light of biological variation. *Nature Reviews Neuroscience* **21**(**7**):384–393
DOI 10.1038/s41583-020-0313-3.

**Wickham H, Averick M, Bryan J, Chang W, McGowan LD, François R, Grolemund
G, Hayes A, Henry L, Hester J, Kuhn M, Pedersen TL, Miller E, Bache SM, Müller
K, Ooms J, Robinson D, Seidel DP, Spinu V, Takahashi K, Vaughan D, Wilke C,
Woo K, Yutani H. 2019.** Welcome to the {tidyverse}. *Journal of Open Source Software*
**4**(**43**):1686 DOI 10.21105/joss.01686.

**Wickham H, François R, Henry L, Müller K. 2020.** dplyr: a grammar of data manipula-
tion. *Available at https://cran.r-project.org/package=dplyr*.

**Yin D, Schwarz EM, Thomas CG, Felde RL, Korf IF, Cutter AD, Schartner CM,
Ralston EJ, Meyer BJ, Haag ES. 2018.** Rapid genome shrinkage in a self-fertile
nematode reveals sperm competition proteins. *Science* **359**(**6371**):55–61
DOI 10.1126/science.aao0827.