# Peer review of "Evolution of fertilization ability in obligatorily outcrossing populations of Caenorhabditis elegans"

_PeerJ, doi:10.7717/peerj.15825_

## Round 0.1 · original submission · Major Revisions

I chose "major revisions" because of the many changes requested by reviewers. My reasons for this decision are as follows: The reviewers' comments and criticisms are constructive and seem reasonable, and they all recommend acceptance after some revisions. It is clear that they want your paper to be in the best possible shape when it appears in print. I do not request any changes other than those requested by the reviewers. Following, I make few remarks regarding the reviewers' comments.

Reviewer #1 asks that you perform a small experiment using at least one obligately outcrossing species such as C. remanei or C. nigoni tested under the same conditions you used with C. elegans, for comparison with your reported experiments. Doing this would strengthen your paper and increase the value of your work significantly.

Reviewer #2 asks you to replace Fig. 2 with a plot of the ratios in Table 2. Then you could put Fig. 1 in a supplemental document. I, too, found Fig. 2 daunting; it shows results for each block clearly, but there are so many blocks I had trouble seeing the overall pattern clearly.

Reviewer #2 also asks if you corrected for multiple testing (statistical testing) in Table 2; see their point 7. If you did not correct for multiple testing, say so, and point out the obvious: your results were clearly not due to random chance alone. Because of your experimental design, you performed 49 statistical tests for mean changes in insemination rate and 49 tests for changes in slope. You were bound to find some statistically significant changes by random chance alone: under a simple chance model with alpha=.05, one expects an average of ~2.5 significant changes in mean and the same number of significant changes in slope. That's not at all what you found. You observed 7 significantly increased means and 2 significantly decreased means, as well as 14 significantly increased slopes and 1 significantly decreased slope. Furthermore, the significant values were disproportionately found in the 12 E populations. Plus, some of the P values were very small (.000). You may wish to combine P values using Fisher's T method. It's easy and quick to do, but you have to decide which tests to combine. Use Fisher's T if you like, but I do not demand that you do it.

If you do make corrections for multiple testing, I suggest (1) that you consult a professional statistician and (2) that you ask her/him about using the Benjamini-Hochberg procedure or the Benjamini-Yekutieli procedure, because the usual methods (e.g., Bonferroni) are too stringent.

Reviewers #1 & #2 remarked that a lack of genetic variation in your starting lines lacked genetic variation probably is the main reason for your getting no response to selection. Please address this briefly in the Discussion.
Reviewer #3 made several excellent suggestions on how to improve the organization of the paper and how to write more clearly. I cannot overemphasize how important this advice is.

Last but not least, I am well aware that IF you add another small experiment, revisions will take you more than the usual time.

Reviewer 1 ·

Basic reporting

The paper is clearly written, with detailed methods that allow replication.
One suggestion for improvement: There is a literature on the genomic bases for the selfing syndrome that might be worth mentioning. In particular, Yin et al. (2018; Science 359:55-61) reported that the self-fertile C. briggsae lost thousands of genes associated with male traits in the short time since it diverged from its outcrossing ancestor. The C. inopinata genome (Kanzaki et al. 2018 Nature Comm. 9: 3216) does not present the same clear evidence of massive “genome shrinkage” in C. elegans, but this may be due to C. inopinata’s peculiar fig niche. Indeed, C. elegans’ transcriptome is less complex and less sex-biased, and its coding gene count smaller, than for other Elegans-group outcrossing species (see Thomas et al. 2012 Current Biology 22:2167).

Experimental design

This was generally good, in terms of controls and replication. However, the lack of genetic variation at the start would have been expected to greatly limit the ability of these populations to respond to selection. This is especially important given the “genome shrinkage” phenomenon noted above: Many loci that historically regulated mating success may now be deleted from the C. elegans genome. Addressing this would require a completely different design, and that is beyond the scope of the MS, but some mention of it in the Discussion is warranted.

Validity of the findings

The findings are carefully described. The works makes clear that there is massive variation in C. elegans mating success. A lingering ambiguity, however, is why it is so variable. I suggest the authors address this with a small-scale control experiment using one or two obligately outcrossing Elegans group species, e.g. C. remanei and/or C. nigoni. By measuring inseminations over time using the exact same conditions, the data can be compared with those already obtained for C. elegans. One predicts there would be much less variation in the outcrossers, and if that is observed it would suggest that the variability observed in C. elegans populations is itself an evolved trait, and not just a limitation of the assay.

Reviewer 2 ·

Basic reporting

Palka et al. describe the results of an experimental evolution study examining the consequences of over a hundred generations of obligate outcrossing in populations derived from self-fertile hermaphrodites. Specifically, they find that insemination ability does not reproducibly change in magnitude and direction across multiple replicates. This manuscript represents a laborious research effort and is a valuable contribution to the literature. Following are my thoughts on how this manuscript could be improved (structured by the PeerJ reviewer guidelines).

1. Data sharing. PeerJ has asked me to check if all underlying data have been provided. The raw data are available on Figshare. However, this should be noted in the manuscript text itself. Beyond this, I could also not find any code describing the statistical analyses (for instance, .R files). Please disregard this comment if I have inadvertently missed this information.

2. Figures. I appreciate the tremendous amount of data that have been collected for this manuscript. However, the data presentation throughout is somewhat overwhelming. For instance, it is challenging for me to synthesize and interpret the information in Figure 2 (with something like >38 panels). One suggestion towards making the data visualization easier to digest would be to plot the ratios in Table 2 instead (and relegate Figure 2 to a supplemental document).

Experimental design

3. Strain construction. I understand this paper is part of a number of manuscripts connected to the same set of experiments. Despite this, further description and justification of the strains used in this paper is needed in order to interpret the results. Specifically, on lines 113-117, the introgression approach is described, but the genetic backgrounds into which the fog-2 mutation was introgressed are not mentioned. What specific genotypes were used for experimental evolution (i.e., isolates/strains/lines with standardized names and citations), and why were they chosen?

Beyond this, I have no other major concerns regarding experimental design.

Validity of the findings

There are no obvious validity or soundness issues with this manuscript. However, I may have a difference of opinion regarding interpretation.

4. The discussion is oriented towards the variation observed among replicates. However, to me, such variation among biological replicates may not be surprising. For many replicates and populations, no significant difference in insemination fraction (or insemination rate) was observed. I think it is worth noting that it is possible there is little standing genetic variation for these traits because these lines are derived from self-fertile hermaphroditic populations. Moreover, de novo mutations that positively impact insemination are also likely to be rare. Thus, it may be unsurprising that consistently reproducible and significant responses to selection on this trait were not observed. The vast background literature on C. elegans experimental evolution, the evolution of reproductive mode, and mutation could also be drawn upon to interpret and contextualize these results (in addition to the material on the replication crisis).

Additional comments

5. Lines 73, 110, & 112. fog-2 is a gene name and should be italicized.

6. Line 115. "...transfers of a single hermaphrodite..." As you introgressed a fog-2 mutation, do you mean females were transferred? The fog-2 mutation causes germ-line feminization in XX animals, preventing self-fertile hermaphroditism.

7. Table 2. I understand an aversion to statistical hypothesis testing is evident in this manuscript, but were the p-values in Table 2 corrected for multiple testing? And, was hypothesis testing applied to the temperature-related results communicated in lines 306-322?

8. Thanks for sharing this work!

·

Basic reporting

In Palka et al the authors use a set of evolved obligatory outcrossing strains of C. elegans to explore the possibility of acquisition of an increase in the rate of male/female productive crosses in a population where the ability for hermaphrodites to self-fertilize has been removed. They use a robust method of having many evolved strains compared to the ancestral strain, replicated in blocks to try an account for variability amongst replicates. They have a number of interesting findings. First, that there is a great amount of variability amongst replicates even with very well controlled experiments. Second, they find candidate lines that may have an increased crossing rate as compared to the ancestral lines. Finally, that there are noticeable effects of temperature the timing and rate of crossing. While all of these findings are exciting and there are no major flaws in the experimentation, there are significant issues with the general structure of the writing, especially the results and discussion section, that make this paper not yet ready to be published (outlined below).


The authors have an interesting question and goal- which they outline well in the introduction and then explain their method in great detail in the Materials and Methods section. However, the results section is poorly organized and difficult to read. If a reader doesn’t read through all the details in the Method section it would completely unclear what was being tested and why. First, the authors need to more clearly outline the particular goals of the experiment in the first part of the results and how they were tested. These can be done briefly and without a lot of detail but with enough information for readers to follow. Most readers will jump to the Results section without reading MM, and maybe not even the Introduction so this is imperative. Second, it would be really helpful if the authors had clear statements of the overall findings found in a paragraph for each paragraph- similar to the first sentence of the second paragraph starting on line 272. For example, the third paragraph is just a list of findings but without any clear overall take home laid out for the reader- even though this paragraph contains one of the three main findings of the paper.

In a similar way the Discussion is not laid out with clarity for the reader. Most Discussions start with a summary of the key findings of the paper and then go on to discuss those findings in relation to the broader scientific field. In this discussion the authors pretty much jump right into a discussion of replication problems- without first placing in context why this was particularly pressing in this experiment and how they saw it here. And while the authors laid out a nice summary of what is known/has been discussed about these issues in the field, they don’t ever clearly explain how they feel their own data is interpreted in the light of these issues. The authors should take a step back and think about the overall structure and flow of the discussion to make it a clearer read.

Experimental design

The experimental design was clearly laid out and appropriate for the questions being asked.

Validity of the findings

The results were appropriately cautiously interpreted and the reasoning behind this laid out. Although as written above- the full logic behind all the conclusions needs to be written out in a more logical manner.

Additional comments

I’m concerned about the use of the term “insemination” as the readout for the experiments. The authors do not directly look at insemination at any point or measure how often insemination occurs without production of progeny (which is a known phenomenon). At a minimum the authors need to explain they are using progeny production as a proxy for insemination events. Clearly, in this system all female that produced progeny experienced a successful insemination event. However, it is feasible that some small number (especially at the elevated temperature) of females that showed zero progeny had experience an insemination event and zero/only a few sperm made it through the uterus to the spermatheca before being expelled. The authors need to either change their term or clearly explain that what they are measuring is a proxy for what they are interested in.

---

## Round 0.2 · accepted · Accept

Please consider replacing or removing Palka et al. and please add the journal to Antol et al. 2022b.
Otherwise, fine.

Reviewer 1 ·

Basic reporting

The paper is clear, and presents a good summary of relevant literature.

Experimental design

No additional concerns, beyond lack of genetic variation in starting population that was noted earlier. Addressing this is beyond the scope of the paper: as written it at least says that new mutations positively impacting mating success are not easily generated over the time scale tested.

Validity of the findings

No additional concerns.

Additional comments

The failure to perform the suggested control experiment is unfortunate, as it makes the study essentially one big negative result. The reason to do it anyway (perhaps via a collaborator) remains: It could tell us if variability per se is an evolved trait (and I strongly suspect it is). However, if the authors truly cannot perform it, it is not essential to meet Peer J's standards.

Reviewer 2 ·

Basic reporting

The authors have addressed my comments.

The authors now explicitly note within the main text where data and code can be retrieved.

Figure 2 is a far more digestible summary of the results.

Experimental design

The starting strains have been clearly stated. I thank the authors for clarifying the methodological approach.

Validity of the findings

The authors now note the lack of standing genetic variation in their starting populations and the implications for experimental evolution.

Additional comments

My only minor comment-- when reading the revised manuscript, I noticed two apparently unpublished cited works. "Palka et al., submitted" is not described in the references. Antol et al. 2022b is listed in the references, but the journal (or other publication type) is not included. I do not know the policies of PeerJ regarding the citation of unpublished works.

·

Basic reporting

No comment

Experimental design

No comment

Validity of the findings

No comment

Additional comments

The authors of Palka et al. "Evolution of fertilization ability in obligatory outcrossing population of Caenorhabditis elegans" made substantial revisions to the writing and analysis of their paper. These revisions make the paper ready for publication. I especially appreciate the changes to Figure 2 and 3 and the clearer flow of writing.